

# Evaluation of time-dependent correlators
# after a local quench in iPEPS: hole motion in the $t-J$ model

Claudius Hubig[1,4][*], Annabelle Bohrdt[2,4], Michael Knap[2,4],
Fabian Grusdt[3,4] and J. Ignacio Cirac[1,4]

**1** Max-Planck-Institut für Quantenoptik, 85748 Garching, Germany
**2** Department of Physics and Institute for Advanced Study,
Technical University of Munich, 85748 Garching, Germany
**3** Department of Physics and Arnold Sommerfeld Center for Theoretical Physics (ASC),
Ludwig-Maximilians-Universität München, 80333 München, Germany
**4** Munich Center for Quantum Science and Technology (MCQST),
80799 München, Germany

★ claudius.hubig@mpq.mpg.de

## Abstract

Infinite projected entangled pair states (iPEPS) provide a convenient variational description of infinite, translationally-invariant two-dimensional quantum states. However, the simulation of local excitations is not directly possible due to the translationally-invariant ansatz. Furthermore, as iPEPS are either identical or orthogonal, expectation values between different states as required during the evaluation of non-equal-time correlators are ill-defined. Here, we show that by introducing auxiliary states on each site, it becomes possible to simulate both local excitations and evaluate non-equal-time correlators in an iPEPS setting under real-time evolution. We showcase the method by simulating the $t-J$ model after a single hole has been placed in the half-filled antiferromagnetic background and evaluating both return probabilities and spin correlation functions, as accessible in quantum gas microscopes.



# 1   Introduction

While tensor network methods in the form of matrix-product states have become the method of choice for the simulation of one-dimensional quantum systems and provide both excellent ground-state data [1] and good accuracy for time-dependent quantities [2], the study of two-dimensional systems remains more difficult. The limited system size of methods such as exact diagonalisation or matrix-product states on a cylinder [3] becomes particularly relevant when studying time-dependent correlators after local excitations, as the system must be able to accommodate the spread of those correlations over time and avoid their interaction with any boundaries. Infinite projected entangled pair states [4–6] (iPEPS) on the other hand allow for the simulation of ground-state properties of *infinite* two-dimensional systems with high accuracy by repeating a finite unit cell of tensors infinitely in both directions. iPEPS were also recently shown to allow for the simulation of global quenches [7–9] at least for short times. This simulation of a real-time evolution following a global quantum quench is relatively straightforward: evolution methods exist [10–12], the quench can be enacted by a change of the Hamiltonian governing this evolution and translational invariance is retained. Equal-time correlators can also be evaluated as usual for each of the computed time-evolved post-quench states.

However, when attempting to simulate a local quench and evaluate non-equal-time correlators, one encounters two problems: First, it is not possible to simply apply an operator (such as $\hat{c}^{\dagger}$) to a single site of the quantum state to create the local excitation: To follow this route, one would have to apply this operator to a specific site, repeated on each unit cell. While making the unit cell itself relatively large is feasible, in this case one merely recovers the case of a finite PEPS calculation and loses the inherent infinity of the iPEPS ansatz. The handling of fermionic commutation rules further complicates this approach.

Second, when pursuing this avenue to simulate the evolution of many excitations – one per unit cell – over time, it is then still not possible to evaluate non-equal-time correlators: These correlators are calculated as expectation values between two different quantum states. However, evaluating the norms of those states will yield either 0 or 1 in the thermodynamic limit and the scale of the correlator is hence not known. In comparison, equal-time correlators are evaluated as $\langle\hat{O}(t)\rangle = \frac{\langle\psi(t)|\hat{O}|\psi(t)\rangle}{\langle\psi(t)|\psi(t)\rangle}$, but the denumerator is clearly ill-defined for a correlator $\langle\hat{O}(t',t)\rangle$ between two different infinite quantum states $\left|\psi(t')\right\rangle$ and $\left|\psi(t)\right\rangle$.

Here, we avoid both problems by adding one auxiliary site to each of the physical sites of our system while preserving translational invariance. We demonstrate the method by evaluating the return probability and diagonal-spin-correlators of a single hole in the two-dimensional antiferromagnetic background of the $t-J$ model [13–23].

# 2   Local excitations and non-equal-time correlators

Consider a system composed of physical local state spaces $\mathcal{H}_i^p$ repeated on each site $i$ of an infinite lattice. We will later focus on the case of a square two-dimensional lattice, but the method likewise applies to other lattice geometries. The total Hilbert space is the tensor product of the local spaces,

$$\mathcal{H}^p = \bigotimes_i \mathcal{H}_i^p \ . \tag{1}$$

We can represent a translationally invariant quantum state $|\psi^p\rangle \in \mathcal{H}^p$ using a tensor network ansatz if it has low entanglement, which is typically true for ground states of local Hamiltonians. If $|\psi^p\rangle$ is only invariant under translation by multiple sites (such as e.g. an antiferromagnetic state under translation by two instead of one site), we can also capture this by using a

sufficiently large unit cell of tensors in the ansatz.

To simulate a local excitation without breaking translational invariance, we now create a translationally invariant superposition of excitations on top of our initial state, simulate the time evolution of this superposition under some Hamiltonian $\hat{H}$ and then select the part of the superposition which contains an excitation at a specific local site [24, 25].

To create the superposition of local excitations, one could apply e.g. $\left(\hat{1} + \epsilon \hat{x}_i^p\right)$ with some creation or annihilation operator $\hat{x}_i^p$ and a small prefactor $\epsilon$ governing the density of excitations on each site as

$$\hat{Y} = \prod_i \left(\hat{1} + \epsilon \hat{x}_i^p\right) . \tag{2}$$

If we let this operator act on our quantum state, we obtain a superposition

$$\hat{Y} |\psi^p\rangle = |\psi^p\rangle + \sum_i \epsilon \hat{x}_i^p |\psi^p\rangle + \mathcal{O}(\epsilon^2). \tag{3}$$

By including a suitable operator (e.g. the particle number operator) in expectation values later, we can select one of the states with an excitation (e.g. a hole at a particular site), which is most likely one of the summands in the second term if $\epsilon$ is small. Crucially, we can also do so after a real-time evolution of $\hat{Y} |\psi^p\rangle$, in this way post-selecting the evolution of a single excitation out of the translationally invariant background.

This approach using $\hat{Y}$ has two downsides: First, the operator $\hat{x}_i^p$ alone typically breaks some symmetry of the system such as spin projection, particle conservation or fermionic parity. While the former two merely lead to a less efficient simulation (as those symmetries then cannot be used in the tensor network ansatz), the breaking of fermionic parity is a serious problem which makes the simulation of fermionic systems impossible. Furthermore, while it is possible to post-select a quantum state with an excitation present at a particular site *after* the time evolution, we cannot post-select for a state where the excitation was *created at a particular site initially*.

To circumvent both problems, we add an auxiliary state space $\mathcal{H}_i^a$ of the same dimension as $\mathcal{H}_i^p$ to each site of our lattice. The total Hilbert space $\mathcal{H}$ is then defined as the tensor product of the auxiliary and physical tensor product spaces on each lattice site

$$\mathcal{H} = \bigotimes_i \left(\mathcal{H}_i^p \otimes \mathcal{H}_i^a\right) . \tag{4}$$

The initial quantum state $|\psi^p\rangle$ is extended by a suitably-chosen empty quantum state $|0^a\rangle$ to form a state in the full Hilbert space $|\psi\rangle = |\psi^p\rangle \otimes |0^a\rangle$. In the case of the $t - J$ model, for example, $|0^a\rangle$ is the state with zero particles on each site in the auxiliary system. The Hamiltonian $\hat{H}$ used for the time evolution still only acts on the physical system.

We then replace the excitation operator $\hat{Y}$ by a form which conserves all symmetries of the system, namely

$$\hat{X} = \prod_i \left(\hat{1} + \epsilon \hat{x}_i^p \left(\hat{x}_i^a\right)^\dagger + \text{h.c.}\right) , \tag{5}$$

where for convenience with existing implementations, we then instead use the local exponential form

$$\hat{X} = \prod_i \exp\left\{\epsilon \hat{x}_i^p \left(\hat{x}_i^a\right)^\dagger + \text{h.c.}\right\} . \tag{6}$$

Instead of creating excitations from nothing as $\hat{Y}$ did, $\hat{X}$ now moves (e.g.) particles from the physical to the auxiliary system and thereby creates an excitation in the physical sector. The density of particles moved and hence the density of local excitations is given by $\epsilon$, ideally we want to consider the case $\epsilon \to 0$. No symmetry is broken during this process if we account for

auxiliary particles in the same way as we account for physical particles and $\hat{X}$ hence leaves the fermionic parity of the state well-defined.

Additionally, it is now possible to not only post-select based on the physical state of some particular site (to select an excitation present there after the evolution), but also to post-select based on the auxiliary state of some particular site. Because there are no dynamics in the auxiliary layer, the auxiliary state at time $t$ is equal to the auxiliary state at time 0 and hence allows for the selection of an excitation which was created at a particular site initially.

## 3 Application to the $t - J$ model

Specifically, we consider the two-dimensional $t - J$ model on the square lattice with a local physical three-dimensional state space $\mathcal{H}_i^p = \text{span}\left\{\left|0_i^p\right\rangle, \left|\uparrow_i^p\right\rangle, \left|\downarrow_i^p\right\rangle\right\}$. Taking a second such space $\mathcal{H}_i^a$ increases the local physical dimension of the iPEPS tensor from three to nine, but iPEPS methods scale favourably in this dimension, so this is not a concern. Let $\hat{c}_{i\sigma}^{p(\dagger)}$ annihilate (create) a physical fermion on site $i$ with spin $\sigma$, let $\hat{s}_i^{p[+,-,z]}$ be the physical spin-$[+,-,z]$ operator on site $i$ (0 if the site is empty) where $\hat{s}^z$ has eigenvalues $\pm 1/2$ and let $\hat{c}_{i\sigma}^{a(\dagger)}$ annihilate (create) an auxiliary fermion on site $i$ with spin $\sigma$. Finally, let $\hat{n}_i^p$ ($\hat{n}_i^a$) denote the particle number operator (0 or 1) on the physical (auxiliary) site $i$.

The Hamiltonian

$$\hat{H} = -t \sum_{\langle i,j \rangle, \sigma} \left( \hat{c}_{i\sigma}^{p\dagger} c_{j\sigma}^p + \hat{c}_{j\sigma}^{p\dagger} c_{i\sigma}^p \right) + J \sum_{\langle i,j \rangle} \left[ \frac{1}{2} \left( \hat{s}_i^{p+} \hat{s}_j^{p-} + \hat{s}_j^{p+} \hat{s}_i^{p-} \right) + \hat{s}_i^{pz} \hat{s}_j^{pz} - \frac{1}{4} \hat{n}_i^p \hat{n}_j^p \right] \tag{7}$$

acts on the physical sector only and is the standard $t - J$ Hamiltonian linking all nearest-neighbour sites $\langle i, j \rangle$. Here, we fix $t = 1$ and $J = 1/3$.

Now take $|\text{GS}\rangle$ to be an approximation of the infinite ground state of $\hat{H}$ at a given iPEPS bond dimension $D$ and half-filling (one fermion per site) in the physical sector, with the auxiliary sector being entirely empty:

$$|\text{GS}\rangle = |\text{GS}^p\rangle \otimes |0^a\rangle . \tag{8}$$

The physical ground state $|\text{GS}^p\rangle$ is simply the ground-state of the Heisenberg Hamiltonian, which can be reasonably well approximated by a $D = 4$ or $D = 5$ iPEPS (other states may of course require a larger bond dimension). This state breaks translational invariance, so we use a $2 \times 2$ unit cell. It preserves both $U(1)_N$ particle number and $U(1)_{S^z}$ spin-projection symmetry and we make use of both [26]. Fermionic commutation relations are ensured using the fermionic tensor network ansatz [27, 28] as implemented in SyTen's STensor class [29, 30].

Given $|\text{GS}\rangle$ as described above, we create the initial excitation with the operator

$$\hat{X} = \prod_i \exp\left\{ \epsilon \sum_\sigma \left( \hat{c}_{i\sigma}^{p\dagger} c_{i\sigma}^a + \hat{c}_{i\sigma}^{a\dagger} c_{i\sigma}^p \right) \right\} . \tag{9}$$

This operator will move particles from the occupied physical sector to the empty auxiliary sector and results in new state $|\psi(0)\rangle$ with a finite hole density on each physical site. Evolving this state under the physical Hamiltonian $\hat{H}$ is straightforward and for a given time $t$ results in a state

$$|\psi(t)\rangle = e^{-it\hat{H}} |\psi(0)\rangle . \tag{10}$$

In the following, we are particularly interested in (a) the return probability $p^R(t)$ of a hole to its creation site and (b) the diagonal spin-spin correlator $z^{\text{diag}}(t)$ at time $t$ with a hole present at time $t$ between the two spins.

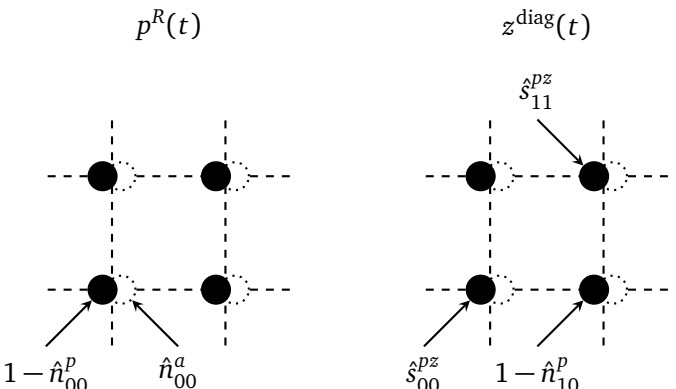

Figure 1: Top view of a single iPEPS unit cell, representing a state $|\psi(t)\rangle$. Each site is the product space of a physical (black) and auxiliary (white/dotted) site. Sites are connected via iPEPS virtual bonds (dashed). Left: The return probability $p^R(t)$ is evaluated by measuring $1-\hat{n}_i^p$ and $\hat{n}_i^a$ at the same iPEPS site. Right: The equal-time correlator $z^{\text{diag}}(t)$ around a hole at time $t$ is evaluated by measuring $\hat{s}_{00}^{pz}$, $\hat{s}_{11}^{pz}$ and $1-\hat{n}_{10}^p$.

The return probability $p^R(t)$ is given by

$$p^R(t) = \frac{\langle\psi(t)|\left(\hat{1}-\hat{n}_i^p\right)\hat{n}_i^a\,|\psi(t)\rangle}{\left\langle\psi(t)\,\middle|\,\hat{n}_i^a\,\middle|\,\psi(t)\right\rangle}\,, \tag{11}$$

where the numerator evaluates the joint probability of a hole created at site $i$ (via the density on the auxiliary site, $\hat{n}_i^a$) present there at a later time (via the density on the physical site, $\hat{n}_i^p$) with the denumerator conditioning on the initial creation of a hole at this site. As the hole density is low, we neglect the case of the hole created at site $i$ moving away and another hole created at some neighbouring site $j$ taking its place.

For the diagonal spin-spin correlator around a hole, let us first define site indices 00, 10 and 11 of the $2 \times 2$ unit cell. The correlator is then

$$z^{\text{diag}}(t) = \frac{\langle\psi|\hat{s}_{00}^{pz}(t)\left(\hat{1}-\hat{n}_{10}^p(t)\right)\hat{s}_{11}^{pz}(t)|\psi\rangle}{\langle\psi|\left(\hat{1}-\hat{n}_{10}^p(t)\right)|\psi\rangle} \tag{12}$$

$$= \frac{\langle\psi(t)|\hat{s}_{00}^{pz}\left(\hat{1}-\hat{n}_{10}^p\right)\hat{s}_{11}^{pz}|\psi(t)\rangle}{\langle\psi(t)|\left(\hat{1}-\hat{n}_{10}^p\right)|\psi(t)\rangle}\,. \tag{13}$$

These correlators are sketched in Fig. 1. Note that, if desired and with larger computational effort, it would be conceivable to repeat the same calculation at different values of $\epsilon$ and subsequently extrapolate $\epsilon \to 0$.

# 4 Results

In the following, we apply the method described above to evaluate the return probability and diagonal-nearest-neighbour spin correlators in the $t-J$ model after the effective introduction of a single hole. We also simulate this system using time-dependent matrix-product states [2] on cylinders of width 4 and 6 to obtain comparison data for short times.

**Time-dependent matrix-product states**  on cylindrical geometries are used to provide comparison data, assumed to be valid at least for short times when the finite circumference of

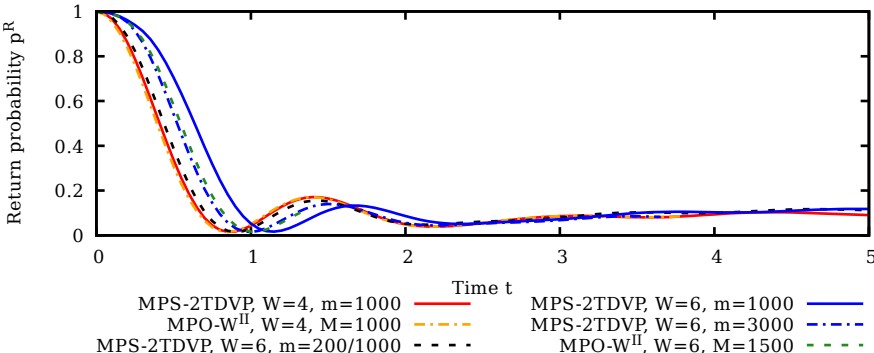

Figure 2: Return probability as calculated using MPS-TDVP or the MPO $W^{\mathrm{II}}$ methods at $J = 1/3$. Both methods used a step size $\delta t = 0.05$. On $W = 4$ cylinders, results are well-converged at $m = 1000$ already. On $W = 6$ cylinders, we only achieve qualitative convergence as the required MPS bond dimension would exceed computational resources.

the cylinders is not yet relevant. We compute the ground-states of the $t - J$ model at half-filling and apply an excitation $\hat{c}_{0,\uparrow} + \hat{c}_{0,\downarrow}$ in the centre of the system. The resulting excited state is then time-evolved with either the 2TDVP [31] or the MPO $W^{\mathrm{II}}$ method [32–34] using the SyTen [29, 30] and TeNPy toolkits [35] respectively. The return probability is given simply as $\langle 1 - \hat{n}_0(t) \rangle$. On cylinders of width $W = 4$, convergence is easy to achieve at modest bond dimensions $m = 1000$, increasing the bond dimension further (up to $m = 5000$) does not lead to different results. As the MPS bond dimension scales exponentially with the circumference of the cylinder, convergence is more difficult on $W = 6$ cylinders. Running the time evolution at the same fixed bond dimension as the initial ground state does not converge well. Preparing the initial ground state at a smaller bond dimension 200 and then running the time evolution at bond dimension $m = 1000$ leads to results at least on short times very similar to the $W = 4$ cylinder (cf. Fig. 2), which is expected as the short-time dynamics are independent of the spin background and hence governed by the hole motion only. Departing from the short-time regime, however, the results become uncontrolled. Increasing the bond dimension further or evolving with the same bond dimension as the initial state does not lead to good convergence. Additionally, while the hole spreads isotropically along the $x$- and $y$-direction on the $W = 4$ cylinder, this is not the case on the $W = 6$ cylinder (not shown). Overall, we only obtain reliable data for the return probability on cylinders of width $W = 4$ and qualitative data for cylinders of width $W = 6$.

**In the iPEPS simulation,** we use the fast full update (FFU, [11, 12]) to obtain the initial ground state and perform the subsequent evolution with the simple update (SU). While the (fast) full update would be able to make better use of the bond dimension of our state, we have encountered some stability issues [8] resulting from this update method which lead to very limited time scales. The simple update may not make perfect use of the iPEPS bond dimension but, given a sufficiently large bond dimension, still provides good results without any of the stability issues observed with the FFU.

We prepare the initial (ground) state at an initial bond dimension $D' = 4$ and create an excitation density of $10^{-2}$. During the subsequent real-time evolution, we allow a range of bond dimensions $D = 4, \ldots, 16$. We focus on even bond dimensions $D$, as odd bond dimensions show slightly worse convergence behaviour due to truncation within spin multiplets. Future computational and algorithmic advances may make bond dimensions $D > 17$ possible. We use a time step size $\delta t = 0.01$ together with a second-order Trotter decomposition of the

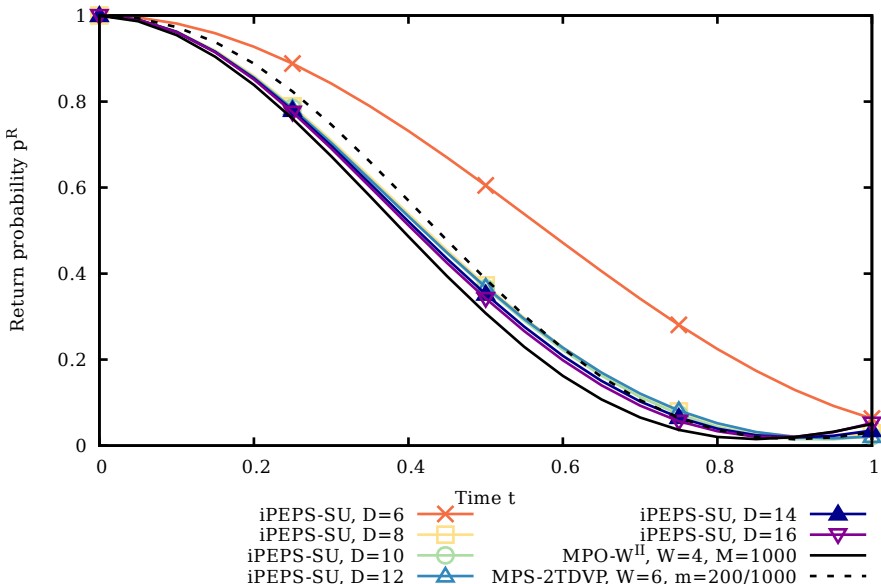

Figure 3: Return probability $p^R(t)$ calculated using iPEPS with the simple update and td-MPS on short times from an initial $D' = 4$ state excited with a global hole density of 0.01 and $J = 1/3$ with various iPEPS bond dimensions $D$. We observe good convergence of the initial decay once $D \geq 8$. Data is evaluated every $\delta t = 0.05$, with symbols shown only for identification.

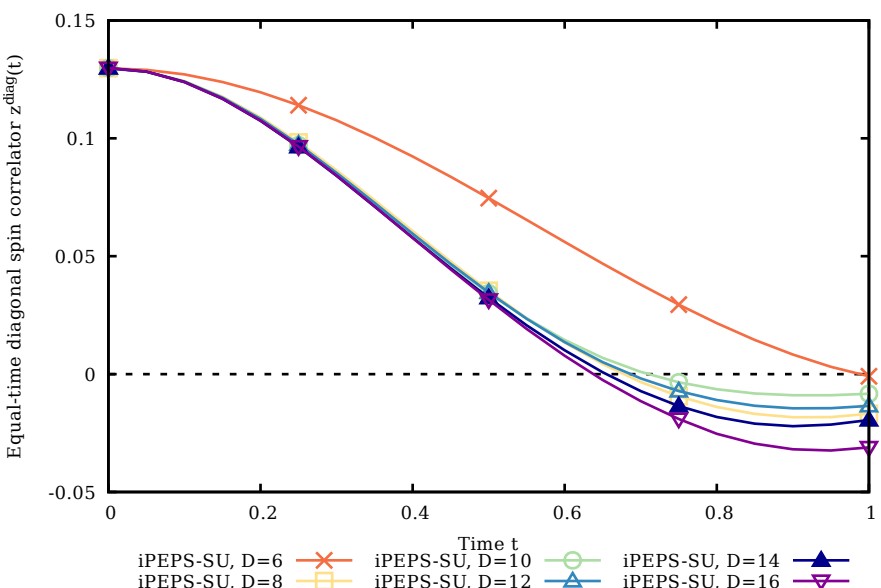

Figure 4: Equal-time diagonal spin correlator $z^{\mathrm{diag}}(t)$ when a hole is present in the lower right side of the two spins calculated using iPEPS with the simple update. The expected zero crossing is observed when increasing the iPEPS bond dimension around time $t \approx 0.6$. Data is evaluated every $\delta t = 0.05$, with symbols shown only for identification.

time-evolution operator.

Exploratory calculations at $D' = 5$ and/or hole density $\approx 10^{-4}$ result in decreased hole mobility at a given evolution bond dimension $D$ as the competition between spin and hole entanglement during the iPEPS state truncation favour the spin sector disproportionally when it is initially more strongly entanglend ($D' = 5$) or there are fewer holes. Hole mobility still increases when increasing the evolution bond dimension $D$, but convergence is much slower than when starting with $D' = 4$.

Expectation values are calculated using the corner transfer matrix at increasing bond dimensions $\chi$ until the difference between results of two successive dimensions $\chi$ and $2\chi$ are sufficiently small; error bars are smaller than symbol sizes in all cases.

Fig. 3 and Fig. 4 show the short-time dynamics of the return probability $p^R(t)$ and diagonal spin-spin correlator $z^{\text{diag}}(t)$ calculated with iPEPS. We observe good convergence in the bond dimension starting from $D \geq 8$ for short times. There, the td-MPS results are reproduced. In particular, the motion of the hole away from its initial site on times of the order of the nearest-neighbour hopping is captured well. At the same time, $z^{\text{diag}}(t)$ becomes negative because the moving hole distorts the original antiferromagnetic background. Hence, spin correlators between both originally nearest-neighbour and originally next-nearest-neighbour fermions contribute to $z^{\text{diag}}(t)$. The stronger nearest-neighbour correlators then dominate the sum and cause the observed sign change. Because the SU(2)-spin symmetry is spontaneously broken along the preferred $z$-axis in the iPEPS calculation but still present in the finite td-MPS calculations, a comparison of numerical values is not meaningful in this case.

For longer times, convergence is very difficult, as our ansatz is inherently limited in entanglement and – due to the simple update – does not make optimal use of the available bond dimension.[1] However, the first revival of the return probability observed in the td-MPS data is still reproduced well by the iPEPS results around $t \approx 1.5$, cf. Fig. 5. The iPEPS data also contains a second, much larger revival at later times $t \approx 3.5$ which is not observed in the td-MPS data and not physically expected either (instead we expect the hole to move away from its creation point with frustrated spins left behind healed by spin flips [18]). At the moment, it is unclear whether this revival is due to limited entanglement in the iPEPS ansatz which hinders healing of frustrated spins through spin-exchange interactions and hence increases the cost of moving the hole further from its origin or a side-effect of the typically overestimated magnetisation in the iPEPS ground state which may lead to more Ising-like physics.

# 5 Conclusion

We have shown that both the simulation of local excitations and the evaluation of time-dependent correlators is possible within the iPEPS formalism. Our predictions, such as the sign-change of diagonal correlators around the hole in Fig. 4, can already be tested in state-of-the-art quantum-gas microscopes [36–39]. Future work using an environment-based truncation scheme such as the FFU together with a stabilised environment (e.g. as introduced in Ref. [40]) will be in a position to make much better use of the available bond dimension than the simple update employed here and hence will be able to analyse the physics of the system for longer times, in particular the interactions between holons and spinons. This would also open an alternative avenue [41] to obtaining spectral functions of two-dimensional systems.

---

[1] A further check on convergence may lie in a deeper analysis of the singular value spectrum obtained after each simple update. While not exact due to missing normalisation of the environment, one might still expect a flattening of the spectrum as entanglement grows over time. We would like to thank Referee 3 for this suggestion.

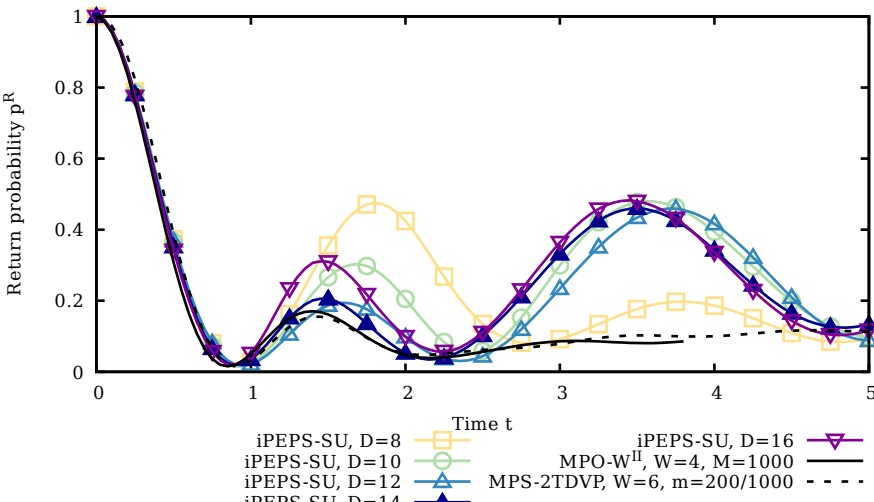

Figure 5: Same as Fig. 3 for longer times $t \geq 1$. The return probability shows qualitative features common to all calculations at large bond dimensions, but quantitative convergence is difficult. The revival around $t \approx 3.5$ is not expected and likely due to limited entanglement in our ansatz.

# Acknowledgements

The authors would like to thank I. Bloch, E. Demler, D. Golez, M. Greiner, I. P. McCulloch, F. Pollmann, and U. Schollwöck for useful discussions.

**Funding information**   C. H. and J. I. C. acknowledge funding through ERC Grant QUENO-COBA, ERC-2016-ADG (Grant no. 742102) by the DFG under Germany's Excellence Strategy – EXC-2111 – 390814868. A.B., F.G., and M.K. acknowledge support from the Technical University of Munich – Institute for Advanced Study, funded by the German Excellence Initiative, the European Union FP7 under grant agreement 291763, the Deutsche Forschungsgemeinschaft (DFG, German Research Foundation) under Germany's Excellence Strategy – EXC-2111 – 390814868, DFG grant No. KN1254/1-1, DFG TRR80 (Project F8), and from the European Research Council (ERC) under the European Union's Horizon 2020 research and innovation programme (grant agreement No. 851161).

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
