# Peer review of "Evaluation of time-dependent correlators after a local quench in iPEPS: hole motion in the t-J model"

_SciPost Physics, doi:SciPost Phys. 8, 021 (2020)_

## Round 1 · Referee Report · Didier Poilblanc · 2019-11-20

Strengths

Cutting-edge developments by experts in the field of tensor networks

Application of the method to an interesting well-studied problem

Weaknesses

Part of the old literature on the one hole t-J model is ignored and may be relevant for interpreting the iPEPS results.

Report

The authors introduce a new method to study, in a two dimensional correlated system, the time evolution after a sudden quench by adapting iPEPS techniques. These are clearly cutting-edge developments by experts in the field of tensor networks.

The authors benchmark the method by investigated the dynamics of a single hole in a quantum antiferromagnet - the so-called t-J model - a well-studied problem in the late 80's - early 90's after the discovery of high-Tc superconductors. Reliable techniques like Lanczos exact diagonalizations on small torii were used to obtain accurately the hole dispersion relation and to establish the composite nature of the hole excitation and the corresponding string-like force. These points were developed in early key references like:

E. Dagotto, R. Joynt, A. Moreo, S. Bacci and E. Gagliano, Phys. Rev. B 41, 9049 (1990).
Didier Poilblanc, H. J. Schulz, and Timothy Ziman, Phys. Rev. B 46, 6435 (1992)
Didier Poilblanc, H. J. Schulz, and Timothy Ziman, Phys. Rev. B 47, 3268 (1993)
D. Poilblanc, T. Ziman, H.J. Schulz and E. Dagotto, Phys. Rev. B 47, 14267 (1993).
P.BĂ©ran, D.Poilblanc and R.B.Laughlin, Nuclear Physics B 473, 707-720 (1996).

The string picture developed in the above references seems to be an essential ingredient to interpret the return probability computed by the authors.

Requested changes

The authors may consider (and cite) the above reference. They may also comment on whether any evidence for the holon-spinon string-like force is seen in their calculation.

  • validity: top
  • significance: top
  • originality: high
  • clarity: high
  • formatting: perfect
  • grammar: excellent

Author:  Claudius Hubig  on 2020-01-07  [id 695]

(in reply to Report 2 by Didier Poilblanc on 2019-11-20)

We are grateful to the referee for their very positive report and are thankful for the additional references suggested.

Of course we are happy to include those additional references once the editor has fixed their editorial recommendation. The time-scales currently available to us do not allow us to compare to the holon-spinon-string picture in detail, although our observations so far are at least consistent with it.

---

## Round 1 · Referee Report · Anonymous · 2019-12-3

Report

In this work the authors present a scheme to compute time-dependent correlation functions based on infinite projected-entangled pair states (iPEPS). Rather than just applying a local operator to the ansatz and time-evolving the state (which would lead to a loss of translational invariance) the scheme keeps the translational invariance of the ansatz by introducing auxiliary states on each site and an operator acting on both the physical and auxiliary space. The accuracy of the operator representation is depends on a small parameter \epsilon (controlling the density of excitations) which is chosen sufficiently small. They test their approach on the t-J model by computing the return probability and diagonal spin correlators of a single hole doped in the antiferromagnetic background at half filling, and they present comparisons with results obtained with matrix product states on cylinders.

Studying local quenches in 2D is a very challenging and important problem. This work introduces a very interesting and useful trick in this context which I believe will be very useful in future iPEPS simulations. As the authors mention, there is still room for improvement in the time evolution scheme used (i.e. a stable full update evolution rather than the less accurate simple update scheme), but even the present results clearly demonstrate the usefulness and applicability of their approach.

For these reasons I can recommend publication of this article in SciPost. The authors may want to take into account the comments and questions listed below when revising their paper.

Comments and questions
(1) In the results section the authors use an excitation density \epsilon of 10^-2, and they mention that ideally one would want to consider the case \epsilon -> 0. While I can imagine that most of the error comes from the truncation due to the finite D, it would nevertheless be good to discuss the dependence of the results on epsilon (e.g. showing some example data as a function of epsilon).

(2) It would be interesting to discuss also the dependence on the trotter step used.

(3) Is there any criterion which can be used in order to judge when the approach starts breaking down? For example, can one observe that at after a certain time scale the singular value spectrum (on the bonds in the simple update scheme) becomes very flat, suggesting that the entanglement is too large / the bond dimension D too small?

  • validity: high
  • significance: high
  • originality: top
  • clarity: good
  • formatting: good
  • grammar: -

Author:  Claudius Hubig  on 2020-01-07  [id 694]

(in reply to Report 3 on 2019-12-03)
Category:
answer to question

We would like to thank the referee for their very positive report and are grateful for their kind comments regarding the applicability and usefulness of our approach.

To reply to their specific questions:

1) We have also tested the simulation with \epsilon of 10^-4 and found fundamentally the same behavior, albeit with a stronger dependence of hole mobility on the bond dimension. That is, to observe hole mobility, we needed a larger bond dimension. This is already discussed in our manuscript (on the lower part of page 7), we expect future work to both study this dependence in more detail and ideally also use smaller \epsilon to avoid hole-hole interactions at longer times.

2) We have found our results not to change if we use a smaller Trotter step size. A larger Trotter step size leads to slightly different results. We expect that if a (fast) full update is used in future work to reduce the truncation error, the Trotter step size gains in relative importance.

3) This is a very interesting remark which we would be happy to explore in future work as well.

---

## Round 2 · Author Response

Dear Editor,

please find attached the resubmission of our manuscript "Evaluation of time-dependent correlators after a local quench in iPEPS: hole motion in the t-J model". We have incorporated the references suggested by the second referee and have also taken into account the suggestion by the third referee.

Kind regards

Claudius Hubig

---

## Round 2 · List of Changes

- include footnote suggesting that the singular value spectrum may provide information on entanglement errors
- include references suggested by the second referee
- include comment in the conclusion that longer times would also allow a closer study of the holon-spinon interaction

---

## Editorial Decision

published